# TabNN: A Universal Neural Network Solution for Tabular Data

## Abstract

Neural Networks (NN) have achieved state-of-the-art performance in many tasks within image, speech, and text domains. Such great success is mainly due to special structure design to fit the particular data patterns, such as CNN capturing spatial locality and RNN modeling sequential dependency. Essentially, these specific NNs achieve good performance by leveraging the prior knowledge over corresponding domain data. Nevertheless, there are many applications with all kinds of tabular data in other domains. Since there are no shared patterns among these diverse tabular data, it is hard to design specific structures to fit them all. Without careful architecture design based on domain knowledge, it is quite challenging for NN to reach satisfactory performance in these tabular data domains. To fill the gap of NN in tabular data learning, we propose a universal neural network solution, called TabNN, to derive effective NN architectures for tabular data in all kinds of tasks automatically. Specifically, the design of TabNN follows two principles: *to explicitly leverage expressive feature combinations* and *to reduce model complexity*. Since GBDT has empirically proven its strength in modeling tabular data, we use GBDT to power the implementation of TabNN. Comprehensive experimental analysis on a variety of tabular datasets demonstrate that TabNN can achieve much better performance than many baseline solutions.

## 1 Introduction

Recent years have witnessed the extraordinary success of Neural Networks (NN), especially Deep Neural Networks, in achieving state-of-the-art performances in many domains, such as image classification (He et al., 2016), speech recognition (Graves et al., 2013), and text mining (Goodfellow et al., 2016). Beside enlarged model capacity, such great achievement of NN is mainly due to the deliberate design of its structures derived from prior knowledge over the certain domain data. For example, Convolutional Neural Networks (CNN) (LeCun et al., 1998) have become the standard solution to address image classification since it can capture the spatial locality by using "Local Receptive Field" (LeCun et al., 1998), which is a common pattern in image data. Recurrent Neural Networks (RNN) (Hochreiter & Schmidhuber, 1997), as another example, has been widely-used on speech recognition and language modeling because its recurrent structure can effectively model the sequential dependency among speech and text data.

In contrast to most of tasks in image, speech, or text domains whose input yields natural spatial or temporal dimension, many other real-world applications, e.g., click through rate prediction (Graepel et al., 2010), time series forecasting (Montgomery et al., 1990; Chatfield, 2000), web search ranking (Agichtein et al., 2006; Cao et al., 2007), etc, bear structured input consisting of multi-dimension meaningful features. Typically, such input data can be generalized as the tabular data, as each row of the tabular corresponds to one data example and each column denotes an individual meaningful feature. Despite the success of CNN and RNN over computer vision, speech recognition, and natural language process, adopting NN over tabular data receives far less attention and yet remains quite challenging. In particular, as illustrated in previous studies (Fernández-Delgado et al., 2014), it usually leads to unsatisfactory performance on tabular data by directly using Fully Connected Neural Network (FCNN), because its fully connected model structure leads to very complex optimization hyper-planes with a high risk of falling into local optimums. Moreover, since different applications usually indicate various effective feature combinations within their respective tabular data, it is quite beneficial to recognize such feature combinations and take advantage of them to design the effective NN model on their tabular data, which however has not been well studied yet.

To address these challenges, we identify two principles for the purpose of designing effective NN models on tabular data: (1) *To explicitly leverage expressive feature combinations*. Rather than blindly pouring all features together into FCNN and learning via back-propagation to discover the implicit feature combinations, it will be beneficial to let NN explicitly leverage the expressive feature combinations. (2) *To reduce model complexity*. Contrary to highly-complex FCNN with too many parameters leading to higher risk of over-fitting or falling into local optimums, it is vital to reduce the complexity of NN models by removing unnecessary parameters and encouraging parameter sharing.

Inspired by these two principles, we propose a universal neural network solution, called TabNN, to derive effective NN architectures for tabular data in all kinds of tasks automatically, by leveraging the knowledge learned by GBDT model (Gradient Boosting Decision Tree) (Friedman, 2001; De'Ath, 2007; Chen & Guestrin, 2016), which has empirically proven its strength in modeling tabular data (Chen & Guestrin, 2016). More specifically, the GBDT-powered TabNN consists of four major steps: (1) *Automatic Feature Grouping (AFG)* automatically discovers feature groups implying effective partial combinations based on GBDT-powered knowledge. (2) *Feature Group Reduction (FGR)* attempts to further cluster feature groups in order to encourage parameter sharing within the same clusters, which can accordingly reduce the complexity of the resulting NN models. (3) *Recursive Encoder with Shared Embedding (RESE)* aims at designing a both effective and efficient NN architecture over clustered tabular feature groups, based on the results of FGR and the feature group importance powered by GBDT. (4) *Transfer Structured Knowledge from GBDT (TSKG)* further leverages structured knowledge within GBDT model to provide an effective initialization for the obtained NN architecture.

To illustrate the effectiveness of the proposed TabNN solution, we conduct extensive experiments on various publicly available datasets with tabular data. Comprehensive experimental analysis has shown that TabNN cannot only create effective NN architectures for various tabular data but also achieves much better performance than other solutions.

In summary, the contributions of this paper are multi-fold:
- We identify two principles for the purpose of designing effective NN models on tabular data.
- We propose TabNN, a general solution for deriving effective NN models for tabular data by leveraging the data knowledge learned by GBDT.
- Extensive experiments show that the proposed method is an off-of-shelf model, which can be ready to use in any kinds of tabular data efficiently and achieves state-of-the-art performance.

## 2 RELATED WORK

**Tabular Data Learning by Tree-Based Models.** Tree-based methods, such as GBDT and Random Forest (Barandiaran, 1998), have been widely applied in many real-world applications, e.g., click through rate prediction (Ling et al., 2017) and web search ranking (Burges, 2010), etc., and have become the first choice in various well-recognized data mining competitions (Chen & Guestrin, 2016). The success of GBDT and other tree-based methods over tabular data majorly relies on their capability on iteratively picking the features with the most statistical information gain to build the trees (Grabczewski & Jankowski, 2005; Sugumaran et al., 2007). Therefore, even if there are amounts of features in the tabular data, GBDT can automatically choose the most useful features to fit the targets well. However, tree-based models still yield two obvious shortages: (1) *Hard to be integrated into complex end-to-end frameworks*. GBDT or other tree-based models cannot back-propagate the error directly to their inputs, thus they cannot be easily plugged into a complex end-to-end framework. To solve this problem, *soft decision trees* or *neural decision trees* have been proposed (Breslow & Aha, 1997; Murthy, 1998; Rokach & Maimon, 2005; Irsoy et al., 2012; Kontschieder et al., 2015; Frosst & Hinton, 2017; Wang et al., 2017) by using differentiable decision functions, instead of non-differentiable axis aligned splits, to construct trees. However, abandoning axis aligned splits will lose the automatic feature selection ability, which is important for learning from tabular data. Feng et al. (2018) propose to use *target propagation* to pass back the error for non-differentiable functions. However, target propagation is inefficient compared with back-propagation as it needs to learn many additional models to propagate the errors. (2) *Hard to learn from streaming data*. Many real-world applications, such as online advertising, continuously generate the large scale of streaming data. Unfortunately, learning tree-based click prediction and recommendation models over streaming data is quite difficult since it usually needs global statistical information to select split points. There have been some works that try to efficiently learn trees from streaming data (Jin & Agrawal, 2003; Gaber et al., 2005; Ben-Haim & Tom-Tov, 2010). However, these models

are specifically designed for the single tree model and their performance cannot achieve the same accuracy as using full data at once. XGBoost (Chen & Guestrin, 2016) and LightGBM (Ke et al., 2017) also provided a simple solution: they learn the structures of trees at first, then, keep the tree structures fixed and update the leaf outputs by the streaming data. Although this solution is simple and efficient, the performance is still worse than learning from all data at once.

**Tabular Data Learning by NN.** These obvious shortages of tree-based methods encourage increasing efforts in applying NN to learn the model over tabular data. Many recent studies attempt to use NN in a variety of applications with tabular data, including the click-through rate prediction (Zhang et al., 2016; Qu et al., 2016; Guo et al., 2017) and recommendation system (Wang et al., 2015; Cheng et al., 2016; Covington et al., 2016; Zhang et al., 2017). Most of them, in fact, focus on how to pre-process categorical features to better adapt to NN. Meanwhile, many numerical features, which are also very important in tabular data, are not well utilized in these works. To sum up, there has no universal NN solution to fit all kinds of tabular data well. The method proposed in this paper aims to fill in this gap and provide an off-of-shelf and universal NN solution.

**Combine NNs with Trees.** Due to the respective pros and cons of NN and tree-based methods, there have been emerging efforts that proposed to combine the NNs and tree-based methods. In general, these efforts can be categorized into two classes: (1) *Tree-like NN*. As pointed by Ioannou et al. (2016), there have been some tree-like NNs, which have decision ability like decision trees to some extent, e.g. GoogLeNet (Szegedy et al., 2015). Rota Bulo & Kontschieder (2014) and Kontschieder et al. (2015) also introduced the tree-like structure and decision ability into NN. However, these works mainly focused on computer vision tasks without attention to tabular data. Yang et al. (2018) proposed the soft binning function to simulate decision trees in NN, which is, however, very inefficient as it enumerates all possible decisions. Wang et al. (2017) proposed NNRF, which used tree-like NN and random feature selection to improve the learning from tabular data. Nevertheless, NNRF simply uses random feature combinations, without leveraging the information from data itself. (2) *Convert Trees to NN*. Another track of works tried to convert the trained decision trees to NNs (Sethi, 1990; Banerjee, 1997; Richmond et al., 2015; Biau et al., 2016; Humbird et al., 2017). However, these works are inefficient as they use a redundant and usually very sparse NN to represent a simple decision tree. When there are many trees, such conversion solution has to construct a very wide NN to represent them, which is unfortunately hard to be applied to realistic scenarios.

**Network Architecture Search.** Apart from converting the tree-based model to NN, other major efforts (Zoph & Le, 2016; Liu et al., 2017; Pham et al., 2018; Luo et al., 2018) proposed to search neural architectures towards a better performance for NNs. However, most of them merely focused on the non-tabular data in computer vision, speech recognition, and natural language process. Particularly, their search space just includes specific structures like convolutional layers or pool layers, which are hardly migrated to the learning from tabular data. Furthermore, such search methods are quite time-consuming they often enumerate the combinations in a large search space.

Given the aforementioned challenges in building or search NN architecture for tabular data, in this paper, we propose an efficient and strategical way to automatically derive effective NN architecture for tabular data, which will be described in details in the following section.

## 3 TABNN

To derive effective NN architecture for tabular data, the design of TabNN follows two key principles: (1) *To explicitly leverage expressive feature combinations*. Rather than blindly pouring all features together into FCNN and learning via back-propagation to discover the implicit feature combinations, it will be beneficial to let TabNN explicitly leverage the expressive feature combinations, meaning that the combination of a certain set of features yields great information gain with respect to the learning task. Compared to learned implicit feature combinations in FCNN, such explicit feature combinations are more robust and can significantly increase the generalization ability of TabNN. (2) *To reduce model complexity*. Too many parameters (i.e. weights or trainable variables) to learn, like FCNN, usually lead to complex optimization hyper-planes so as to result in a high risk of over-fitting. Therefore, to improve the efficiency as well as the effectiveness of learned NN model, it is critical for TabNN to reduce the complexity of designed NN architecture by removing the unnecessary parameters and encouraging parameter sharing.

In this paper, based on these two principles, we propose a GBDT-powered TabNN. Specifically, as shown in Alg. 1, TabNN contains four major steps: (1) *Automatic Feature Grouping* (AFG,

Line 2-3): to follow the first principle, this step automatically discovers the effective feature groups (i.e. expressive feature combinations) from a tabular dataset $\mathbb{D}$ by leveraging GBDT. Therefore, the designed NN model can explicitly leverage the feature combinations derived from feature groups. We employ $G$ to stand for the set of all feature groups. The cardinal of $G$, i.e. the number feature groups, may be very large as GBDT often requires many trees to achieve good performance. (2) *Feature Group Reduction* (FGR, Line 4): AFG may produce many feature groups and therefore results in too many parameters. To reduce the parameters and encourage the parameter sharing as guided by the second principle, we cluster these feature groups into $k$ sets, i.e. $G_1, \cdots, G_k$, based on the similarity over these feature groups. Since there are common features over the clustered feature groups, we can leverage this characteristic to significantly reduce the parameters, by reusing the embedding of these common features in the derived architecture. (3) *Recursive Encoder with Shared Embedding* (RESE, Line 6): we design a both effective and efficient NN architecture over clustered tabular feature groups, based on the results of FGR step and the feature group importance powered by GBDT. (4) *Transfer Structural Knowledge from GBDT* (TSKG, Line 8-10): beside feature grouping knowledge, trees in GBDT also contain rich structural knowledge. This step aims at transferring the structural knowledge in GBDT to the obtained NN architecture.

In the rest of this section, we will dive into more details of these steps one by one.

| **Algorithm 1:** TABNN | **Algorithm 2:** GREEDY FGR |
|---|---|
| 1   Donate the dataset as $\mathbb{D}$, and the number of feature group sets as $k$ | 1   Initialize all sets $G_1, \cdots, G_k$ to be empty set |
| 2   Trees $T \leftarrow$ TRAINGBDT$(\mathbb{D})$ | 2   Initialize hyper-parameters $n$ and $\alpha$ |
| 3   $G \leftarrow \bigcup_{t_i \in T}$ FEATUREGROUP$(t_i)$ | 3   **for** $l \leftarrow 1 : n$ **do** |
| 4   **execute** GREEDY FGR algorithm with $G$ | 4     $G_j^{(l)} \leftarrow \emptyset, \forall 1 \leq j \leq k$ |
| 5   **for** $j \leftarrow 1 : k$ **do** | 5     Randomly initialize a processing order $\pi(G)$ |
| 6     Construct RESE module for $G_j$ with $\boldsymbol{\theta}_j$ | 6     **foreach** feature group $g$ **in** $\pi(G)$ **do** |
| 7     $T_j \leftarrow$ TREESET$(G_j)$ | 7       $j \leftarrow \arg\max_i \left( \left\| G_i^{(l)} \cup \{g\} \right\|_\alpha - \left\| G_i^{(l)} \right\|_\alpha \right)$ |
| 8     Leaf prediction $\mathbb{L}_j \leftarrow$ PREDLEAF$(\mathbb{D}, T_j)$ | 8       $G_j^{(l)} \leftarrow G_j^{(l)} \cup \{g\}$ |
| 9     Leaf embedding $\mathbb{H}_j \leftarrow$ EMB$(\mathbb{L}_j)$ | 9     $i \leftarrow \arg\max_l \min_j \left\| G_j^{(l)} \right\|_\alpha$ |
| 10   Use $(\mathbb{D}, \mathbb{H}_j)$ to initialize $\boldsymbol{\theta}_j$ | 10    $G_j \leftarrow G_j^{(i)}, \forall 1 \leq j \leq k$ |
| 11   Assemble $k$ RESE modules and train it with $\mathbb{D}$ | |

**Automatic Feature Grouping.** The AFG component is designed under the guidance of the first principle to determine which expressive feature combinations should be explicitly utilized by TabNN. Since different tabular data may indicate various expressive feature combinations, it is inappropriate to recognize a predefined static feature grouping for all kinds of tabular data. Therefore, it is necessary to design a dynamical and automatic approach to identify important feature combinations for tabular data. Although many popular methods, such as correlation test, feature clustering, principal component analysis, etc., can be applied to obtain the feature groups dynamically, they fail to identify expressive complex combination among features within the same group. On the other hand, the tree-based models provide a goldmine for discovering rich non-linear dependencies among features (Sugumaran et al., 2007). Specifically, those features within one tree is indeed a well-processed feature group with rich expressiveness. Inspired by that, it becomes quite natural to use the tree-based model to automatically find expressive feature groups. Among various options of tree-based method, in this paper, we adopt GBDT for two major reasons: first, GBDT has been widely used to model tabular data of many real-world applications; moreover, as gradient boosting will adjust the learning targets for different trees as latest residuals, GBDT can learn many diverse trees such that it can create many diverse feature groups.

More formally, suppose the set of trees trained in the GBDT model is $T$, we will use the features within the same tree $t \in T$ as a feature group $g \in G$. Since a GBDT often contains many trees to achieve good performance, there will be many feature groups. The characteristics of GBDT have decided that such feature groups can have many overlapping features, which enable us to merge these feature groups into much compact sets to reduce the complexity by parameter sharing in TabNN.

**Feature Group Reduction.** To find similar feature groups for parameter sharing, FGR is designed to merge all feature groups into $k$ sets with the minimum number of common features in one set maximized. More formally, the objective of the FGR is to maximize the value of $\min_{1 \leq j \leq k} |\bigcap_{g \in G_j} g|$, where $|\cdot|$ stands for the number of features in the set. Indeed, there are two major challenges to

address FGR. The first one is the computational complexity, i.e., the FGR problem is NP-hard. In fact, we have following theorem:

**Theorem 1.** *The NP-hard $Pm||C_{\max}$ schedule problem can be reduced to the FGR problem.*

*Proof.* (Sketch.) In the well known $Pm||C_{\max}$ problem (Lawler et al., 1993), there are $n$ jobs to be scheduled on $m$ identical machines. Each job $j$ has a process time $p_j$. The objective is to design a schedule plan to minimize the max load $C_{\max}$ of all machines. To show the hardness of FGR problem, we proof that any instance of $Pm||C_{\max}$ problem can be reduced to a instance of FGR problem. W.l.g., we suppose all the process time are integers and their summation is $N$, i.e. $N = \sum_{j=1}^{n} p_j$. Now we set $k = m$ and construct an instance of FGR problem. We define the total feature sets as $F = \{1, 2, \cdots, N\}$. The $j$-th feature group $g_j = F \setminus \{N_{j-1} + 1, \cdots, N_j\}$, where $N_j = \sum_{i=1}^{j} p_i$. These feature groups satisfy the following property: the size of $g_{j_1} \cap g_{j_2} \cap \cdots \cap g_{j_t}$ is exactly $N - p_{j_1} - \cdots - p_{j_t}$. According to this property, we can find that minimizing the load on one machine is equivalent to maximizing the intersection of corresponding feature groups. Thus, the $Pm||C_{max}$ problem can be reduced to the FGR problem, which means the FGR problem is harder than the $Pm|C_{max}$ problem. Consequently, the FGR problem is NP-hard. $\square$

Another challenge lies in that, with the increasing number of feature groups, their intersections usually keep shrinking, which unfortunately makes it hard to share weights for similar feature groups. To address this challenge, we adopt *soft intersections* instead of the origin one. We define the *$\alpha$-soft intersection* as the set of features which are covered by $\alpha$ fraction of all feature groups. For convenience, the operator $\| \cdot \|_{\alpha}$ is used to calculate the size of $\alpha$-soft intersection of feature groups. As the origin FGR is a special case with $\alpha = 1$, thus, the soft intersection version of FGR is also an NP-hard problem. Due to the NP-hardness of this problem, it is impossible to compute an optimal solution efficiently even when $k$ is given in advance. Thus, we take a heuristic approach for the efficiency purpose. Our algorithm is shown in Alg. 2. In this algorithm, we enumerate all feature groups in a random order and add it greedily into the feature group set with the greatest gain (Line 7 and 8). This procedure will be repeated $n$ times and the one with largest minimum $\alpha$-soft intersection will be selected as the final sets of feature groups (Line 9 and 10). Although such a simple solution cannot guarantee an optimal solution, it is very efficient and can provide a sub-optimal solution.

**Recursive Encoder with Shared Embedding.** After FGR generates $k$ sets of feature groups, it still remains challenging to organize many feature groups within a single set into an efficient NN architecture. Fortunately, the resulting sets of FGR yield two characteristics that can inspire an efficient design. In particular, the first one corresponds to diverse importance of different feature groups within one set due to the varying importance of trees in GBDT, and such important difference can stimulate a more efficient recursive NN architecture to let more important feature groups have the more direct impact on the task. Furthermore, the second characteristic correlates to many common features within one resulting set of FGR caused by the $\alpha$-soft intersection, and such common features can share parameters for the purpose of efficient learning when constructing the NN architecture.

Inspired by these two characteristics, we propose a *recursive encoder with shared embedding (RESE)* approach to constructing NN architectures based on feature group sets generated by FGR. The whole RESE architecture is summarized in Fig. 1, in which the circles stand for the neurons and all arrows stand for fully connections to these neurons; and, for convenience, we use smaller indices to represent the layers closer to the output layer.

As shown in this figure, RESE takes advantage of a recursive NN architecture to allow more important feature groups to contribute more directly on the task. In particular, we first reorder the feature group in each set according to the descending importance of corresponding trees, and define the $i$-th important feature group in set $G_j$ as $G_{j,i}$. Then, we arrange more important group as the input of layers closer to the output. To further reduce the model complexity, RESE is designed to exponentially increase the number of feature groups in layer $l_i$ along with increasing $i$. For example, RESE can put $G_{j,1}$ (the most important feature group in $G_j$) as input in layer $l_1$, $\{G_{j,2}, G_{j,3}\}$ in layer $l_2$, $\{G_{j,4}, \cdots, G_{j,7}\}$ in layer $l_3$, and so on. In this way, the designed recursive architecture will be quite compact since it has reduced the number of layers logarithmically.

To encourage using sharing parameters for common features within each feature group set, we first extract the common features from $G_j$ and concatenate them as a vector $\hat{x}_j$ and use it with corresponding embedding component (green circles) as shared input to all layers. More specifically, as illustrated in Fig. 1, the input of each layers consists of the common features $\hat{x}_j$ with their embedding components, external feature vector (for layers except $l_0$), and the output of previous layer (the

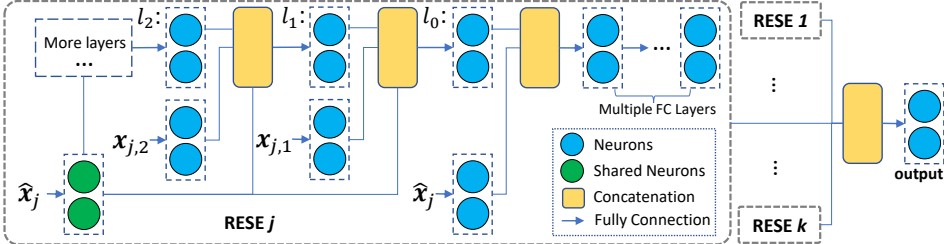

Figure 1: Architecture of TabNN, which contains $k$ Recursive Encoder with Shared Embedding (RESE) modules. The embedding of common features $\hat{\boldsymbol{x}}_j$ (green blocks) is shared in these layers.

layer with larger index). For layers $l_i$ with $1 \leq i \leq \lceil \log_2(|G_j|+1) \rceil$, the external feature vector $\boldsymbol{x}_{j,i}$ is a concatenation of features in $\{G_{j,p} \cup G_{j,p+1} \cup \ldots \cup G_{j,2p-1}\} \setminus \hat{\boldsymbol{x}}_j$, for $p = 2^{i-1}$. Note that the size of $\boldsymbol{x}_{j,i}$ is small in practice as the common features are excluded.

In summary, the designed architecture arranges the feature groups in an efficient way by leveraging their importance. Moreover, as shown in Fig. 1, the embedding of common features $\hat{\boldsymbol{x}}_j$ are reused. By employing share embedding, we can not only reduce the number of parameters but also result in more efficient back-propagation over common features in the deeper layers.

REMARK: For the completeness of content without disturbing the elaboration of the main design, we organize more details in this remark part. (1) *Shrink of feature representation:* for the purpose of information extraction and parameters reduction, the output dimension of each layer is set to be a fraction of the dimension of its input. Specifically, this fraction is set to $0.25$ for raw feature inputs (such as $\hat{\boldsymbol{x}}_j$ and $\boldsymbol{x}_{j,i}$), and to $0.5$ for other inputs (such as the output of concatenation components). (2) *Non-linear activation:* for all neurons, we use batch normalization (Ioffe & Szegedy, 2015) following by a ReLU (Nair & Hinton, 2010) as the non-linear activation. (3) *Multiple FC layers before output:* to enhance the expressiveness ability, there are multiple fully connected layers, which is defined by hyper-parameter, between layer $l_0$ and the output layer. (4) *Final combination:* as shown in Fig. 1, all outputs of $k$ RESE modules will be concatenated as the inputs of a final fully connected layer.

**Transfer Structural Knowledge from GBDT.** Besides the knowledge of expressive feature combination, the GBDT model also contains rich structural knowledge which is quite invaluable to further improve the learning efficiency and model effectiveness. In this paper, we adopt the knowledge distillation technology (Hinton et al., 2015) to transfer GBDT's structural knowledge into more effective model initialization for TabNN.

Formally, we define corresponding trees of $G_j$ in set $j$ as an ordered set $T_j$, and $T_{j,i}$ represents the $i$-th important tree in $G_j$. To transfer structural knowledge of these trees, we first use the training data $\mathbb{D}$ to go through these trees one by one. For each data sample, we will get a set of leaf indices output by trees in $G_j$. As the categorical data are hard to handle by NN, we extend these indices with one-hot representation and denote it as a vector $\boldsymbol{L}_{j,d}$ for the $d$-th sample in $\mathbb{D}$. Let $\mathbb{L}_j$ stand for the one-hot vectors for all data samples. In fact, the pairwise data $(\mathbb{D}, \mathbb{L}_j)$ can sufficiently represent the structural knowledge over the training data, since different data samples will go through the different paths in trees according to their feature values and finally reach a leaf node. This part is corresponding to the Line 8 in the Alg. 1.

However, as there are many leaf node in the tree set $T_j$, the dimension of $\boldsymbol{L}_{j,d}$ could be very high. Thus, learning from it could be extremely inefficient. So we adopt embedding technology (Mikolov et al., 2013) to reduce the dimension while retain important information in $\boldsymbol{L}_{j,d}$. To speed up the embedding learning, rather than using the unsupervised AutoEncoder (Bengio et al., 2009) method, we use a FCNN with one hidden layer to learn the embedding supervised. More specifically, based on bijection relations between leaf indices and leaf values, the one-hot coding $\boldsymbol{L}_{j,d}$ of leaf index is taken as input, while the corresponding leaf value is taken as the training label. Then the output of the hidden layer is the embedding of $\boldsymbol{L}_{j,d}$, which is defined as $\boldsymbol{H}_{j,d}$. We denote the whole embedding set as $\mathbb{H}_j$. The Line 9 in Alg. 1 is corresponding to this part. After $\mathbb{H}_j$ is prepared, we use the data $(\mathbb{D}, \mathbb{H}_j)$ to pre-train the parameter $\boldsymbol{\theta}_j$ of the RESE module corresponding to $G_j$. After all RESE modules are initialized, we can concatenate these modules together and normally train the whole architecture from the ground truths.

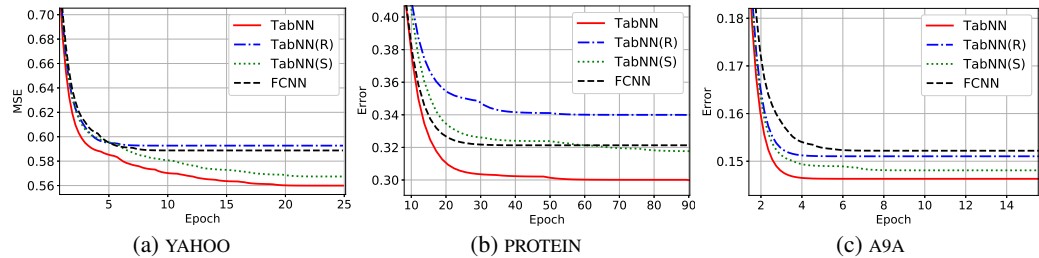

Figure 2: Epoch-Metric Curves[2].

Table 1: Details of datasets used in experiments.

| Name | #Train | #Test | #Feature | Task | Metric |
|---|---|---|---|---|---|
| YAHOO | 0.47M | 0.17M | 699 | Regression | MSE |
| LETOR | 2.27M | 0.75M | 136 | Regression | MSE |
| PROTEIN | 14.8K | 6.62K | 357 | Classification | Error |
| A9A | 32.6K | 16.3K | 123 | Classification | Error |
| FLIGHT | 7.29M | 0.50M | 52 | Classification | Error |

Table 2: The #parameter in LETOR data when deal with many trees (feature groups). "Concat" refer to the concatenation of multiple FCNNs, each of which models from one feature group.

| #Trees | 20 | 50 | 100 | 500 | 1000 |
|---|---|---|---|---|---|
| Concat | 268K | 670K | 1.34M | 6.7M | 13M |
| RESE | 154K | 175K | 197K | 240K | 261K |

## 4 EXPERIMENTS

In this section, we conduct thorough evaluations on TabNN[1] by comparing its performance with several baseline methods over a couple of public tabular datasets. Through the experiments, we mainly demonstrate the advantages of TabNN over the tabular data with numerical features and low-cardinality categorical ones, since high-dimension and sparse data, on the other hand, has been well modeled by existing Deep-and-Wide NN methods Cheng et al. (2016). The basic information of the used tabular datasets are listed in Table 1 and more details can be found in A.1. We can find that these datasets cover diverse real-world applications. To ensure an efficient learning of NN over tabular features, we normalize all numerical features and convert categorical features to numerical vectors by tool "categorical-encoding" (Scikit-learn, 2018) with "binary" and "leave-one-out" encoding.

In the following experiments, we compare TabNN with the following baselines: (1) *GBDT* is a widely used tree-based learning algorithm for modeling tabular data. (2) *FCNN* is the traditional NN solution for tabular data. To achieve the best performance, we use NNI (Microsoft, 2018b) to search the best hyper-parameter settings. (3) *NRF (GBDT)* (Biau et al., 2016) converts the regression trees to NN. For the fair comparison with other baselines, we convert from GBDT to NN, rather than Random Forest (Barandiaran, 1998). As NRF (GBDT) only work for regression tasks, we cannot compare its performance on some classification tasks. (4) *NNRF* (Wang et al., 2017) is a recent NN solution for the tabular data. As NNRF is only designed for classification tasks, we cannot compare with it on all datasets. Since these baselines are introduced in Sec. 2, we do not provide more details about them in this section due to the space restriction.

To further evaluate the effectiveness of knowledge brought from GBDT, we add two simplified variants of TabNN for comparison: (1) *TabNN (R)* randomly clusters features into several groups without using any knowledge from GBDT; (2) *TabNN (S)* only uses structural tree knowledge while keeps using random feature combinations. Except for FCNN, we did not search the hyper-parameters for TabNN and other baselines. And for TabNN, we set the learning rate as $0.001$, $k = 10$, and $\alpha = 0.5$ in all experiments. More setting details are available in A.2.

**Overall Performance Comparison.** Table 3 compares the performance of TabNN with all baselines on five tabular datasets. From this table, we can find that TabNN outperforms all other baselines on all datasets. In particular, even though we did not search the hyper-parameters for TabNN, it is still much better than well-tuned FCNN. A further comparison of training curves between TabNN and FCNN, as shown in Fig. 2, illustrates that TabNN convergences much faster than FCNN. Moreover, though both NRF (GBDT) and TabNN can leverage the knowledge learned by GBDT, TabNN can further improve the performance and outperform GBDT, while NRF (GBDT) will be over-fitting when continued training the NN converted from GBDT. In addition, while we have tried many settings to fine-tune NNRF, the performance of it remains even worse than FCNN. Since NNRF paper

---
[1]The codes will be released to GitHub after acceptance.

[2]Due to space restriction, we put the curves of rest two datasets in Appendix A.3.

Table 3: Overall comparison on test data. The results are run by 5 different random seeds.

|  | YAHOO | LETOR | PROTEIN | A9A | FLIGHT |
|---|---|---|---|---|---|
| GBDT | 0.5700 | 0.5493 | 0.3031 | 0.1475 | 0.1826 |
| FCNN | 0.5905 ±1e-3 | 0.5569 ±4e-4 | 0.3358 ±3e-3 | 0.1531 ±2e-3 | 0.1850 ±7e-4 |
| NRF (GBDT) | 0.6010 ±9e-3 | 0.5665 ±2e-3 | N/A | N/A | N/A |
| NNRF | N/A | N/A | 0.5066 ±3e-3 | 0.1869 ±2e-3 | 0.1932 ±1e-5 |
| TabNN (R) | 0.5955 ±2e-3 | 0.5543 ±4e-4 | 0.3407 ±3e-3 | 0.1571 ±2e-3 | 0.1866 ±1e-3 |
| TabNN (S) | 0.5711 ±2e-3 | 0.5478 ±5e-4 | 0.3187 ±3e-3 | 0.1488 ±1e-3 | 0.1799 ±1e-3 |
| TabNN | **0.5612 ±8e-4** | **0.5461 ±5e-4** | **0.3022 ±2e-3** | **0.1473 ±6e-4** | **0.1764 ±3e-3** |

Table 4: Comparison on streaming data learning, on FLIGHT data.

| Batch | 1 | 2 | 3 | 4 | 5 | 6 | 7 |
|---|---|---|---|---|---|---|---|
| GBDT | 0.2399 | 0.2383 | 0.2245 | 0.1567 | 0.1745 | 0.1646 | 0.2655 |
| GBDT with leaf updates | 0.2395 | 0.2335 | 0.2182 | 0.1569 | 0.1648 | 0.1585 | 0.2679 |
| TabNN | **0.2344** | **0.2297** | **0.2135** | **0.1563** | **0.1607** | **0.1527** | **0.2598** |

only reports experiment results on small datasets (less than ten thousand samples), we hypothesize that NNRF may work well for small datasets but not suitable for large datasets as used in our experiments. To summarize, all these experimental results demonstrate that TabNN can outperform all kinds of baselines and achieve superior performance in tabular data learning.

**Analysis of TabNN.** In this part, we further investigate the importance of key components in TabNN. The performance gap between TabNN (S) and TabNN, as shown in Table 3, indicates that feature grouping knowledge from GBDT plays a vital role in TabNN. Similarly, the comparison between TabNN (S) and TabNN (R), as shown in Table 3, implies that the structural knowledge from GBDT also yields a large contribution to the performance of TabNN. Besides the gaps in final performance, tree knowledge also boosts TabNN with faster convergence, as shown in Fig. 2. Obviously, these results reveal the importance of tree knowledge brought by GBDT in TabNN. To disclose the importance of RESE module in parameters reduction, Table 2 shows the number of parameters used for one feature group set with varying number of trees (i.e. feature groups). The basic approach *Concat* simply concatenate many FCNNs, each of which is learned based on one feature group. From this table, we can find that the parameters in *Concat* increases linearly with growing number of trees, while RESE can significantly reduce the size of parameters logarithmically. Thus, RESE module indeed plays an important role to reduce the model complexity. To sum up, these results demonstrate that the key components in TabNN are indispensable to enhance the effectiveness and efficiency in tabular data learning.

**Compared with GBDT in Streaming Data Learning.** As mentioned in Sec. 2, one shortage of GBDT is the difficulty in learning from streaming data. To demonstrate the advantage of TabNN in streaming data, we design a simulation experiment based on FLIGHT data. Specifically, we use the data samples of the first 4 months in the year 2007 to train an initial model, and then update it once a month, i.e. one batch contains the data corresponding to one month. On the other hand, we train two GBDT baselines: one is only trained by the data of first 4 months without further learning from steaming data, while another one will continued use streaming data to update leaf values ('refit' function in LightGBM (Microsoft, 2018a)). As shown in Table 4, GBDT without the sequential update is the worst as expected, while TabNN achieves the best results in streaming data learning. Furthermore, this table also implies that, with using more data, TabNN can give rise to more significant performance improvement. These results demonstrate the advantage of TabNN in streaming data learning.

## 5 CONCLUSION

To fill the gap of NN in tabular data learning, we propose a universal neural network solution, called TabNN, which can derive the effective neural architectures automatically for tabular data. The design of TabNN follows two principles, one as explicitly leveraging expressive feature combinations and the other as reducing model complexity. Since GBDT is proven to be effective in tabular data, we leverage GBDT to power the implementation of TabNN. Specifically, TabNN first leverages GBDT to automatically identify expressive feature groups and then clusters feature groups into sets to encourage parameter sharing. After that, TabNN utilizes tree importance knowledge from GBDT to construct recursive NN architectures. To enhance the training efficiency and learning performance, tree structural knowledge is also utilized to provide an effective initialization for the derived architecture. Extensive experiments on various tabular datasets show the advantages of TabNN in modeling tabular data and demonstrate the necessity of designed components in TabNN.

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

## APPENDIX A    MORE DETAILS IN EXPERIMENTS

### A.1    DATASET DETAILS.

The information of the selected tabular datasets are listed in Table 1. Among these datasets, YAHOO (Chapelle & Chang, 2011) and LETOR (Liu et al., 2007) are both the learning-to-rank datasets, and we solve them as regression problems (i.e. the pointwise ranking problems). PROTEIN (Wang, 2002) and A9A (Kohavi, 1996; Platt, 1999) are the classification datasets, which are downloaded from LIBSVM website (Chang & Lin, 2011). And we use FLIGHT (Expo, 2009) as a classification data by forecasting the flights are delayed (more than 15 minutes) or not. Specifically, we use the data in the year 2007 for the training and the 50k samples in the year 2008 for the test. For the data without test sets, we randomly sample 80% for the training, and use the rest for the test.

### A.2    MODEL DETAILS

The concrete model setting for all used models is listed in Table 5.

Table 5: Model Setting Details.

| Model | Setting Details |
|---|---|
| GBDT | For all the experiments, we use the released LightGBM (Microsoft, 2018a) with learning rate 0.15 and strict 100 trees even when meeting multi-class classification dataset. Besides, the leaf number is set to $0.5 \times$ #feature on most datasets (except 0.4 on A9A), and is limited to the range $[32, 128]$ to avoid underfitting or overfitting. By the way, the GBDT used in all the other models is set as same. |
| FCNN | We use batch normalization and ReLU as activation, and AdamW (Loshchilov & Hutter, 2017) as optimizer for FCNN. As for the hyper-parameters and structures setting, we use NNI (Microsoft, 2018b) toolkit to run various models with different settings 64 times on each dataset, then select the best one among all models to be our baseline. The hyper-parameter searching in NNI contains learning rate, batch size and FCNN structure (the number of layers and corresponding hidden units). The searched best model on each dataset is listed in Table 6, all of which outperform the human setting one. |
| NRF(GBDT) | Based on the author's released code (JohannesMaxWel, 2018), we change the Random Forest with GBDT in NRF for the fair comparison. Besides, considering the tree dependency in GBDT, we train the whole sparse neural network converted by NRF at once. |
| NNRF | According to the original paper (Wang et al., 2017), we set the depth of NNRF to $\lceil \log_2(\text{#class}) \rceil + 1$, set the sampled input feature dimension to $\sqrt{\#feature}$ and use 150 NNs with bootstrapping for ensemble. |
| TabNN
TabNN(R)
TabNN(S) | In all experiments, we set the learning rate to 0.001, batch size to 128, optimizer to AdamW, $k = 10$, $\alpha = 0.5$ and add three hidden layers with 200, 100, 50 units correspondingly before the 20-dimension output of RESE module. |

Table 6: The best setting of FCNN on each dataset.

| Dataset | Learning rate | Batch size | Inputs | Hidden units | Outputs |
|---|---|---|---|---|---|
| YAHOO | 1.87e-3 | 128 | 699 | [1048, 524, 262, 131] | 1 |
| LETOR | 6.58e-4 | 128 | 136 | [204, 163, 130, 104, 83] | 1 |
| PROTEIN | 1.89e-3 | 64 | 357 | [535, 374] | 3 |
| A9A | 2.53e-4 | 128 | 123 | [92, 64, 44] | 2 |
| FLIGHT | 8.47e-3 | 128 | 52 | [78, 70, 63] | 2 |

### A.3    MORE EPOCH-METRIC CURVES

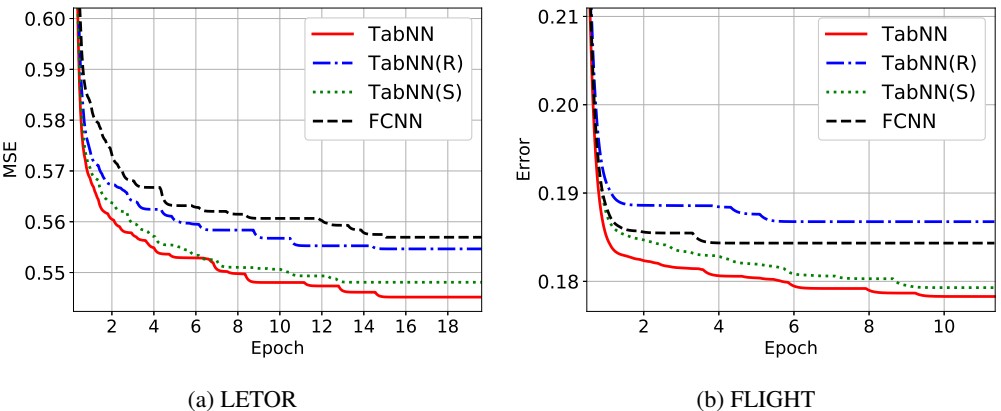

(a) LETOR                 (b) FLIGHT

Figure 3: More Epoch-Metric Curves.

