# OpenReview forum: "TabNN: A Universal Neural Network Solution for Tabular Data"
_ICLR.cc/2019/Conference_

### Official Review · AnonReviewer3 · 2018-11-02
**Interesting idea**

**Rating:** 5
**Confidence:** 2

**Review:**

The paper proposed an interesting algorithm and direction, which tries fill the gap of NN in tabular data learning. My concern is, given this is an empirical work,  the number of datasets used in evolution is a bit small.

Also, xgboost was the winning algorithm for many competitions for tabular data, would be good to compare the NN with properly optimised xgboost.

In chapter 2, related work. The authors state that "tree-based models still yield two obvious shortages: (1) Hard to be integrated into complex end-to-end frameworks... (2) Hard to learn from streaming data.

To me these two reasoning statements are not particularly convincing. One could also say:

NN models yield two obvious shortages: (1) Hard to be integrated into complex end-to-end frameworks... (2) Hard to learn from streaming data...

Actually, tree ensemble based algorithms, eg Hoeffding tree ensembles, are among the best performed algorithms for data streaming tasks.

---

> ### Author Response · Authors · 2018-11-07
> **Response to the Comments of Reviewer 3**
>
>
> Thanks for your efforts in reviewing our paper and the valuable comments. We attempt to address your concerns using the following points and hope they can help you better understand our work.
>
> 1. the number of benchmark datasets
>
> Currently, there are 5 datasets in our experiments. Actually, we have evaluated the proposed methods by conducting experiments in many datasets and observed the similar results. Due to the space restriction, however, we cannot present them all in the paper. We can provide more experiment results in appendix to eliminate this concern.
>
> 2. XGBoost
>
> We use LightGBM to learn GBDT model in the experiment part. LightGBM is proven comparable (even better) with XGBoost in many Kaggle competitions (refer to https://www.kaggle.com/shivamb/data-science-trends-on-kaggle and https://github.com/Microsoft/LightGBM/tree/master/examples). Therefore, we think using LightGBM is sufficient for comparison.
>
> 3. Two shortages of tree-based models
>
> Let us describe these two shortages with more details here.
>
>   1) Hard to be integrated into complex end-to-end frameworks. In such framework, there are many modules, each of which may correspond to one sub-task with a global optimization goal. The outputs of modules can serve as the inputs of other modules. Therefore, to train such a framework in an end-to-end way, the module should be able to propagate the errors from its outputs to its inputs. NN can naturally support this, as its learning algorithm is the back-propagation. In contrast, tree-based models do not support this as the tree learning process is not differentiable and therefore cannot propagate the errors to its inputs. As stated in the Section 2, although there are some works targeting to address this problem, these solutions will lose the automatic feature selection ability and cannot work well on the tabular data.
>
>   2) Hard to learn from streaming data. For NN's learning, we can use Stochastic Gradient Descent (SGD) or mini-batch SGD to naturally learn from streaming data, since the NN model could be updated per data sample or per mini-batch of samples. However, it is not effective for tree-based model to support this as its learning needs the global statistical information. Using the partial statistical information may produce the sub-optimal split points and results in worse models. There are some works addressed this problem, like Hoeffding trees, which stores the statistical histograms into leaf nodes.
>   However, most of these solutions are designed for the single decision tree. Although there are ensemble versions of them, most of them are based on bagging (like Random Forest), which is proven not as good as GBDT.
>
> In short, NN does not suffer from these two problems due to its mini-batch back-propagation learning process. In contrast, tree-based model is hard to solve these two problems due to its learning algorithm is based on global statistical information. Therefore, TabNN is a better general solution for tabular data.

---

> > ### Public Comment · (anonymous) · 2018-11-08
> > **Hoeffding trees do not cache training examples**
> >
> > It is incorrect to say that Hoeffding trees cache training examples. They maintain a histogram of class--feature value combinations in each leaf node, which is independent of the number of examples observed. In practice, this does not require very much memory. For example, try training a Hoeffding tree using the MOA package [1] on the popular HIGGS dataset---often used for benchmarking gradient boosting frameworks. You will see that, with the default hyperparameters, the model size barely exceeds 30MB.
> >
> > [1] https://moa.cms.waikato.ac.nz/

---

> > > ### Author Response · Authors · 2018-11-08
> > > **Thanks for clarifying, we will update the comment accordingly.**
> > >
> > > Yeah, you are right. Sorry for drawing the wrong conclusion and thanks for pointing it out.

---

### Official Review · AnonReviewer1 · 2018-11-05
**An applied paper addressing a significant problem and research direction but missing the novel foundations getting to the bottom of the problem.**

**Rating:** 4
**Confidence:** 5

**Review:**

This paper proposes a hybrid machine learning algorithm using Gradient Boosted Decision Trees (GBDT) and Deep Neural Networks (DNN). The intended research direction on tabular data is essential and promising. However, the proposed technique does not seem to be handling the problem foundationally well. It seems heavily dependent on GBDT. It also shows itself in the results that final algorithm is almost indistinguishable from GBDT regarding results. Moreover, I  don't think that the data sets in experiments are good enough to cover the importance and the nature of the problem.

Pros:
-This is a crucial line of research direction that aims to make DNNs applicable to many real-world problems (beyond speech and vision) in which discrete data and heterogeneous features exist such as engagement prediction, recommendation, and search.
-The starting point of using GBDT seems like a good choice.
-The Paper is mostly well written except occasional repetitions and missing acronym definitions.

Cons:
-The proposed technique does not seem to be original enough, and it does not handle the problem foundationally well. I do not think that there is enough justification/demonstration for the fact that a general NN solution for Tabular Data invented. The proposed technique is heavily dependent on GBDT (Indeed the algorithm and the learned trees are used at least three times). This shows itself in the results; i.e., the proposed algorithm is either negligibly performing better than GBDT or when  GBDT dependence removed, it performs worse. It seems to me that (except the minor small section of streaming data), the paper is more like a proper verification of how tree-based learning algorithms work very well in tabular data--which is far from the basis of the paper and does not make the paper novel enough for ICLR.
-The proposed technique seems to include very heavy feature engineering and several ad-hoc practical steps--that is far from the motivation of using NN in tabular data.
-In the provided benchmark data sets the depth of the analysis seems to be enough. However, in the proposed domain of tabular data, often data sets are significantly more high dimensional in reality and include at least one set of sparse large dimensional features  (e.g., unstructured raw text for the search queries.) In such scenarios, it had been showed that wide-and-deep NNs perform decently. However such problems are entirely missing in the results section. I also think that this is a lost opportunity for the authors as they could be showing that it is the NN part contributing.

---

> ### Public Comment · (anonymous) · 2018-11-07
> **Cannot agree with AnonReviewer1**
>
> The reviewer 1 said this paper didn't handling the problem foundationally well and therefore was not novel, for its heavily dependencies on GBDT. This opinion is obviously not correct.
>
> Firstly, handling the problem foundationally is not equal to novelty. There are many novelties in all kinds of directions, e.g. ideas, solutions, technique, etc.
>
> Secondly, it is hard to define which solutions are foundationally well, and why depending on GBDT is not foundationally well? NN should be standalone without helps of other models? NN has been evolving, are there any reasons to block us to leverage GBDT into NN?
>
> Disclaimer: I just disagree with AnonReviewer1's opinions about ”handling the problem foundationally well“

---

> ### Author Response · Authors · 2018-11-07
> **Response to the Comments of Reviewer 1**
>
>
> Thanks for your efforts in reviewing our paper and the valuable comments, but we have different opinions about your comments.
>
> 1. Comments about the contributions and novelty
>
> As we emphasized many times in our paper, the success of DNN in domains such as image, speech and text, is built on the comprehensive exploration of the locality-based patterns, which motivates us to first find such patterns of features in tabular data automatically and then build up NN architecture based on these discovered patterns. This is the core idea of this paper. Thus, GBDT is just a tool we adopt to mine the patterns and do feature grouping since GBDT is an efficient and convenient method for these pre-processing tasks: 1) GBDT is very fast. In most experiments, the total time cost of GBDT part in TabNN is about several minutes, while the NN part often needs several hours for training. 2) the learning of GBDT is just based on statistical information over full dataset. Thus, GBDT can learn the stable and robust feature combinations.
> We can definitively replace GBDT with other methods, such as feature correlations, as long as they can achieve better performance then GBDT.
>
> Regarding the comments asking for the comparison with GBDT, we consider that they are not comparable since we are not inventing a model to beat GBDT, instead, we are developing a model to cover the scenarios not suitable for GBDT such as some applications need online updating. This point has also been emphasized in our paper.
>
>
> 2. Heavy feature engineering and ad-hoc practical steps
>
> We are not sure why you conclude this point. TabNN is a fully end-to-end learning approach with no need of an extra feature engineering step.
> And as stated in the paper, the design of TabNN follows two principles: \emph{to explicitly leverages expressive feature combinations} and \emph{to reduce model complexity}. We cannot agree there are ad-hoc parts in the proposed model. Could you explain this with more details?
>
>
> 3. Benchmark Dataset and Compared with Deep and Wide (D&W) NNs
>
> As stated in Section 2, D&W NNs and many related models can work well with high dimensional sparse features, which are usually in the form of one-hot encoding converted from categorical features. Actually, these NNs perform very well in such datasets, even better than GBDT.
>
> In contrast, the proposed TabNN works better on another kinds of tabular data, with numerical features and low-cardinality categorical features. Since there are many dummy dimensions in one-hot encoding, TabNN is hard to learn the useful features combinations from them.
>
> Therefore, TabNN and D&W NNs are orthogonal with each other. We can use them independently according to the feature types of data. And they can be used together for the data with mixed feature types.
>
> Therefore, we did not conduct any experiment on data with high-cardinality categorical features. We will state this clearer in the paper.

---

### Official Review · AnonReviewer2 · 2018-11-08
**Review of submission 585**

**Rating:** 5
**Confidence:** 4

**Review:**

Summary: This paper introduces a new Neural Network training procedure, designed for tabular data, that seeks to leverage feature clusters extracted from GBDTs.

Strengths: The idea of leveraging feature groups in a neural network structure; the novelty of the RESE model;

Weaknesses: The main weakness of the paper is that the performance gains are extremely low compared to the next contender; perhaps they are statistically significant (this cannot be determined), but it's unclear why we wouldn't use GBDT.

Minor typos:
(abstract)
- "NN has achieved" => "Neural Networks have achieved"
- "performances" => performance
- "explicitly leverages" => "explicitly leverage"

Questions:
- (top of p. 2) What exactly is the difference between "implicit feature combinations" and "explicit (?), expressive feature combinations"
- (top of p. 2) "encourage parameter sharing" - between what and what? at which level? [reading on, I realized this applies to groups of features; it should maybe be made clear earlier]
- what is the benefit brought by the 'Structural Knowledge' transfer? is this quantified anywhere? based on the description, I don't understand if this is an add-on to TabNN or whether it is incorporated in TabNN.


Recommendations for the authors: Would it be possible to provide an analysis of the cases when TabNN is expected to outperform GBDT by a sizable margin? Or, if not, are there other reasons why using a neural network would make more sense than just simply running GBDT?

---

> ### Author Response · Authors · 2018-11-08
> **Response to the Comments of Reviewer 2**
>
> Thanks for your efforts in reviewing our paper and the valuable comments. We attempt to address your concerns in the following.
>
> 1. Response to the "Weaknesses" part and the comparison with GBDT
>
> As stated in the response to review 1, our goal is not inventing a model to beat GBDT but developing a model to cover the scenarios not suitable for GBDT such as some applications need online updating.
>
> "The next contender" model in your comment is the GBDT, which indeed works well for tabular data. However, GBDT suffers from two shortages, as stated in Section 2 and the responses to reviewer 3. These 2 shortages make GBDT very hard to be used in many real-world scenarios. For example, in an online recommender system, we need to update the model frequently to achieve the satisfying real-time performance. In this case, GBDT will be very inefficient as it needs to be re-trained from scratch. In contrast, NN can be learned by mini-batch fashion and therefore can learn from streaming data naturally.
>
> The proposed TabNN can overcome these shortages and achieve comparable accuracy with GBDT. Moreover, compared with previous NN based solutions for tabular data, TabNN outperforms them significantly. Therefore, TabNN is a better general solution for tabular data as it can cover more scenarios.
>
>
> 2. Difference between "implicit feature combinations" and "explicit feature combinations"
>
> The main difference lies in whether the feature combination information is explicitly introduced into model structure or not. For example, in FCNN, as all features are connected to the neurons in the next layer, there are no feature combination information in the model structure. Although the feature combination information are not explicitly provided, one neuron in FCNN can learn a linear combination of its input features. Thus, we say there are "implicit feature combinations" in FCNN.
>
> In TabNN, we leverage GBDT to find feature combinations and then construct model structure according to them. Thus, we say there are "explicit feature combinations" in TabNN.
>
> "Implicit feature combinations" is not efficient as it introduces much more trainable parameters, and has a risk of over-fitting. In contrast, "explicit feature combinations" let model focus on the more important feature combinations and is more efficient. The successful CNN model also uses "explicit feature combinations", as it only combines the local pixels.
>
>
> 3. About "encourage parameter sharing".
>
> Yes, we use parameter sharing in the one cluster of feature groups. We will clarify this in the paper.
>
> 4. Benefits brought by the "Structural Knowledge"
>
> We had compared the benefit brought by the 'Structural Knowledge' in the experiment. The difference between TabNN (S) and TabNN (R), as shown in Table 3, implies that that the structural knowledge from GBDT yields a large contribution to the performance of TabNN.
>
> The "Structural Knowledge" is in TabNN by default. We will clarify this in the paper.

---

### Meta-Review · Area_Chair1 · 2018-12-02
**Reject**

**Confidence:** 4
**Recommendation:** Reject

**Metareview:**

All reviewers agree in their assessment that this paper has merits but is not yet ready for acceptance into ICLR. The area chair commends the authors for their responses to the reviews.